# Optimal Shrinkage of Singular Values Under Random Data Contamination

**Danny Barash**
School of Computer Science and Engineering
Hebrew University
Jerusalem, Israel
danny.barash@mail.huji.ac.il

**Matan Gavish**
School of Computer Science and Engineering
Hebrew University
Jerusalem, Israel
gavish@cs.huji.ac.il

## Abstract

A low rank matrix $X$ has been contaminated by uniformly distributed noise, missing values, outliers and corrupt entries. Reconstruction of $X$ from the singular values and singular vectors of the contaminated matrix $Y$ is a key problem in machine learning, computer vision and data science. In this paper, we show that common contamination models (including arbitrary combinations of uniform noise, missing values, outliers and corrupt entries) can be described efficiently using a single framework. We develop an asymptotically optimal algorithm that estimates $X$ by manipulation of the singular values of $Y$, which applies to any of the contamination models considered. Finally, we find an explicit signal-to-noise cutoff, below which estimation of $X$ from the singular value decomposition of $Y$ must fail, in a well-defined sense.

## 1   Introduction

Reconstruction of low-rank matrices from noisy and otherwise contaminated data is a key problem in machine learning, computer vision and data science. Well-studied problems such as dimension reduction [3], collaborative filtering [24, 28], topic models [13], video processing [21], face recognition [35], predicting preferences [26], analytical chemistry [29] and background-foreground separation [4] all reduce, under popular approaches, to low-rank matrix reconstruction. A significant part of the literature on these problems is based on the singular value decomposition (SVD) as the underlying algorithmic component, see e.g. [7, 19, 23].

Understanding and improving the behavior of SVD in the presence of random data contamination therefore arises as a crucially important problem in machine learning. While this is certainly a classical problem [14, 17, 20], it remains of significant interest, owing in part to the emergence of low-rank matrix models for matrix completion and collaborative filtering [9, 34].

Let $X$ be an $m$-by-$n$ unknown low-rank matrix of interest ($m \leq n$), and assume that we only observe the data matrix $Y$, which is a contaminated or noisy version of $X$. Let

$$Y = \sum_{i=1}^{m} y_i \mathbf{u}_i \mathbf{v}_i' \tag{1}$$

be the SVD of the data matrix $Y$. Any algorithm based on the SVD essentially aims to obtain an estimate for the target matrix $X$ from (1). Most practitioners simply form the Truncated SVD (TSVD) estimate [18]

$$\hat{X}_r = \sum_{i=1}^{r} y_i \mathbf{u}_i \mathbf{v}_i' \tag{2}$$

where $r$ is an estimate of $rank(X)$, whose choice in practice tends to be ad hoc [15].

Recently, [10, 16, 32] have shown that under white additive noise, it is useful to apply a carefully designed *shrinkage* function $\eta : \mathbb{R} \to \mathbb{R}$ to the data singular values, and proposed estimators of the form

$$\hat{X}_\eta = \sum_{i=1}^{n} \eta(y_i)\mathbf{u}_i\mathbf{v}_i' . \tag{3}$$

Such estimators are extremely simple to use, as they involve only simple manipulation of the data singular values. Interestingly, in the additive white noise case, it was shown that a unique optimal shrinkage function $\eta(y)$ exists, which asymptotically delivers the same performance as the best possible rotation-invariant estimator based on the data $Y$ [16]. Singular value shrinkage thus emerged as a simple yet highly effective method for improving the SVD in the presence of white additive noise, with the unique optimal shrinker as a natural choice for the shrinkage function. A typical form of optimal singular value shrinker is shown in Figure 1 below, left panel.

Shrinkage of singular values, an idea that can be traced back to Stein's groundbreaking work on covariance estimation from the 1970's [33], is a natural generalization of the classical TSVD. Indeed, $\hat{X}_r$ is equivalent to shrinkage with the *hard thresholding* shrinker $\eta(y) = \mathbf{1}_{y \geq \lambda}$, as (2) is equivalent to

$$\hat{X}_\lambda = \sum_{i=1}^{n} \mathbf{1}_{y_i \geq \lambda}\mathbf{u}_i\mathbf{v}_i' \tag{4}$$

with a specific choice of the so-called *hard threshold* $\lambda$. While the choice of the rank $r$ for truncation point TSVD is often ad hoc and based on gut feeling methods such as the Scree Plot method [11], its equivalent formulation, namely hard thresholding of singular values, allows formal and systematic analysis. In fact, restricting attention to hard thresholds alone [15] has shown that under white additive noise there exists a unique asymptotically optimal choice of hard threshold for singular values. The optimal hard threshold is a systematic, rational choice for the number of singular values that should be included in a truncated SVD of noisy data. [27] has proposed an algorithm that finds $\eta^*$ in presence of additive noise and missing values, but has not derived an explicit shrinker.

## 1.1 Overview of main results

In this paper, we extend this analysis to common data contaminations that go well beyond additive white noise, including an arbitrary combination of additive noise, multiplicative noise, missing-at-random entries, uniformly distributed outliers and uniformly distributed corrupt entries.

The primary contribution of this paper is formal proof that there exists a unique asymptotically optimal shrinker for singular values under uniformly random data contaminations, as well a unique asymptotically optimal hard threshold. Our results are based on a novel, asymptotically precise description of the effect of these data contaminations on the singular values and the singular vectors of the data matrix, extending the technical contribution of [16, 27, 32] to the setting of general uniform data contamination.

**General contamination model.** We introduce the model

$$Y = A \odot X + B \tag{5}$$

where $X$ is the target matrix to be recovered, and $A, B$ are random matrices with i.i.d entries. Here, $(A \odot B)_{i,j} = A_{i,j}B_{i,j}$ is the Hadamard (entrywise) product of $A$ and $B$.

Assume that $A_{i,j} \overset{\text{iid}}{\sim} (\mu_A, \sigma_A^2)$, meaning that the entries of $A$ are i.i.d drawn from a distribution with mean $\mu_A$ and variance $\sigma_A^2$, and that $B_{i,j} \overset{\text{iid}}{\sim} (0, \sigma_B^2)$. In Section 2 we show that for various choices of the matrix $A$ and $B$, this model represents a broad range of uniformly distributed random contaminations, including an arbitrary combination of additive noise, multiplicative noise, missing-at-random entries, uniformly distributed outliers and uniformly distributed corrupt entries. As a simple example, if $B \equiv 0$ and $P(A_{i,j} = 1) = \kappa$, then the $Y$ simply has missing-at-random entries.

To quantify what makes a "good" singular value shrinker $\eta$ for use in (3), we use the standard Mean Square Error (MSE) metric and

$$L(\eta|X) = \left\|\hat{X}_\eta(Y) - X\right\|_F^2 .$$

Using the methods of [16], our results can easily be extended to other error metrics, such as the nuclear norm or operator norm losses. Roughly speaking, an optimal shrinker $\eta^*$ has the property that, asymptotically as the matrix size grows,

$$L(\eta^*|X) \leq L(\eta|X)$$

for any other shrinker $\eta$ and any low-rank target matrix $X$.

The design of optimal shrinkers requires a subtle understanding of the random fluctuations of the data singular values $y_1, \ldots, y_n$, which are caused by the random contamination. Such results in random matrix theory are generally hard to prove, as there are nontrivial correlations between $y_i$ and $y_j$, $i \neq j$. Fortunately, in most applications it is very reasonable to assume that the target matrix $X$ is low rank. This allows us to overcome this difficulty by following [15, 27, 32] and considering an asymptotic model for low-rank $X$, inspired by Johnstone's Spiked Covariance Model [22], in which the correlation between $y_i$ and $y_j$, for $i \neq j$ vanish asymptotically.

We state our main results informally at first. The first main result of this paper is the existence of a unique asymptotically optimal hard threshold $\lambda^*$ in (4).

Importantly, as $\mathbb{E}(Y) = \mu_A X$, to apply hard thresholding to $Y = A \odot X + B$ we must from now on define

$$\hat{X}_\lambda = \frac{1}{\mu_A} \sum_{i=1}^{n} \mathbf{1}_{y_i > \lambda} \mathbf{u}_i \mathbf{v}_i' .$$

**Theorem 1.** *(Informal.) Let $X$ be an $m$-by-$n$ low-rank matrix and assume that we observe the contaminated data matrix $Y$ given by the general contamination model* (5). *Then there exists a unique optimal (def. 3) hard threshold $\lambda^*$ for the singular values of $Y$, given by*

$$\lambda^* = \sigma_B \sqrt{\left(c + \frac{1}{c}\right)\left(c + \frac{\beta}{c}\right)}$$

*where $\beta = m/n$ and $c = \sqrt{1 + \beta + \sqrt{1 + 14\beta + \beta^2}}/\sqrt{2}$.*

Our second main result is the existence of a unique asymptotically optimal shrinkage function $\eta^*$ in (equation (3)). We calculate this shrinker explicitly:

**Theorem 2.** *(Informal.) Assume everything as in Theorem 1. Then there exists a unique optimal (def. 3) shrinker $\eta^*$ for the singular values of $Y$ given by*

$$\eta^*(y) = \begin{cases} \dfrac{\sigma_B^2}{y\mu_A} \sqrt{\left(\left(\dfrac{y}{\sigma_B}\right)^2 - \beta - 1\right)^2 - 4\beta} & y \geq \sigma_B(1 + \sqrt{\beta}) \\ 0 & y < \sigma_B(1 + \sqrt{\beta}) \end{cases}$$

We also discover that for each contamination model, there is a critical signal-to-noise cutoff, below which $X$ cannot be reconstructed from the singular values and vectors of $Y$. Specifically, let $\eta_0$ be the zero singular value shrinker, $\eta_0(y) \equiv 0$, so that $\hat{X}_{\eta_0}(Y) \equiv 0$. Define the *critical signal level* for a shrinker $\eta$ by

$$x^{critical}(\eta) = \inf_x \left\{ x : L(\eta|X) < L(\eta_0|X) \right\}$$

where $X = x\tilde{\mathbf{u}}\tilde{\mathbf{v}}'$ is an arbitrary rank-1 matrix with singular value $x$. In other words, $x^{critical}(\eta)$ is the smallest singular value of the target matrix, for which $\eta$ still outperforms the trivial zero shrinker $\eta_0$. As we show in Section 4, a target matrix $X$ with a singular value below $x^{critical}(\eta)$ cannot be reliably reconstructed using $\eta$. The critical signal level for the optimal shrinker $\eta^*$ is of special importance, since a target matrix $X$ with a singular value below $x^{critical}(\eta^*)$ cannot be reliably reconstructed using *any* shrinker $\eta$. Restricting attention to hard thresholds only, we define $x^{critical}(\lambda)$, the critical level for a hard threshold, similarly. Again, singular values of $X$ that fall below $x^{critical}(\lambda^*)$ cannot be reliably reconstructed using *any* hard threshold.

Our third main result is the explicit calculation of these critical signal levels:

**Theorem 3.** *(Informal.) Assume everything as in Theorem 1 and let $c$ be as in Theorem 1. Let $\eta^*$ be the optimal shrinker from Theorem 2 and let $\lambda^*$ be the optimal hard threshold from Theorem 1. The critical signal levels for $\eta^*$ and $\lambda^*$ are given by:*

$$
\begin{aligned}
x^{critical}(\eta^*) &= (\sigma_B/\mu_A) \cdot \beta^{\frac{1}{4}} \\
x^{critical}(\lambda^*) &= (\sigma_B/\mu_A) \cdot c.
\end{aligned}
$$

Finally, one might ask what the improvement is in terms of the mean square error that is guaranteed by using the optimal shrinker and optimal threshold. As discussed below, existing methods are either infeasible in terms of running time on medium and large matrices, or lack a theory that can predict the reconstruction mean square error. For lack of a better candidate, we compare the optimal shrinker and optimal threshold to the default method, namely, TSVD.

**Theorem 4.** *(Informal.) Consider $\beta = 1$, and denote the worst-case mean square error of TSVD, $\eta^*$ and $\lambda^*$ by $M_{TSVD}$, $M_{\eta^*}$ and $M_{\lambda^*}$, respectively, over a target matrix of low rank $r$. Then*

$$
\begin{aligned}
M_{TSVD} &= \left(\frac{\sigma_B}{\mu_A}\right)^2 5r \\
M_{\eta^*} &= \left(\frac{\sigma_B}{\mu_A}\right)^2 2r \\
M_{\lambda^*} &= \left(\frac{\sigma_B}{\mu_A}\right)^2 3r.
\end{aligned}
$$

Indeed, the optimal shrinker offers a significant performance improvement (specifically, an improvement of $3r(\sigma_B/\mu_A)^2$, over the TSVD baseline.

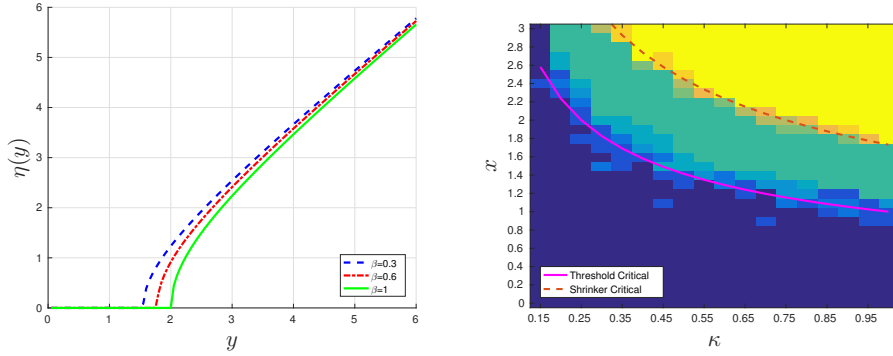

Figure 1: Left: Optimal shrinker for additive noise and missing-at-random contamination. Right: Phase plane for critical signal levels, see Section 6, Simulation 2.

Our main results allow easy calculation of the optimal threshold, optimal shrinkage and signal-to-noise cutoffs for various specific contamination models. For example:

1. **Additive noise and missing-at-random.** Let $X$ be an $m$-by-$n$ low-rank matrix. Assume that some entries are completely missing and the rest suffer white additive noise. Formally, we observe the contaminated matrix

$$
Y_{i,j} = \begin{cases} X_{i,j} + Z_{i,j} & \text{w.p. } \kappa \\ 0 & \text{w.p. } 1-\kappa \end{cases},
$$

where $Z_{i,j} \overset{\text{iid}}{\sim} (0, \sigma^2)$, namely, follows an unknown distribution with mean 0 and variance $\sigma^2$. Let $\beta = m/n$. Theorem 1 implies that in this case, the optimal hard threshold for the singular values of $Y$ is

$$
\lambda^* = \sqrt{\sigma^2 \kappa \, (c + 1/c) \, (c + \beta/c)}
$$

where $c = \sqrt{1 + \beta + \sqrt{1 + 14\beta + \beta^2}}/\sqrt{2}$. In other words, the optimal location (w.r.t mean square error) to truncate the singular values of $Y$, in order to recover $X$, is given by $\lambda^*$. The

optimal shrinker from Theorem 2 for this contamination mode may be calculated similarly, and is shown in Figure 1, left panel. By Theorem 4, the improvement in mean square error obtained by using the optimal shrinker, over the TSVD baseline, is $3r\sigma^2/\kappa$, quite a significant improvement.

2. **Additive noise and corrupt-at-random.** Let $X$ be an $m$-by-$n$ low-rank matrix. Assume that some entries are irrecoverably corrupt (replaced by random entries), and the rest suffer white additive noise. Formally,

$$Y_{i,j} = \begin{cases} X_{i,j} + Z_{i,j} & \text{w.p. } \kappa \\ W_{i,j} & \text{w.p. } 1 - \kappa \end{cases} .$$

Where $Z_{i,j} \overset{\text{iid}}{\sim} (0, \sigma^2)$, $W_{i,j} \overset{\text{iid}}{\sim} (0, \tau^2)$, and $\tau$ is typically large. Let $\tilde{\sigma} = \sqrt{\kappa\sigma^2 + (1-\kappa)\tau^2}$. The optimal shrinker, which should be applied to the singular values of $Y$, is given by:

$$\eta^*(y) = \begin{cases} \tilde{\sigma}^2/(y\kappa)\sqrt{\left((y/\tilde{\sigma})^2 - \beta - 1\right)^2 - 4} & y \geq \tilde{\sigma}(1 + \sqrt{\beta}) \\ 0 & y < \tilde{\sigma}(1 + \sqrt{\beta}) \end{cases} .$$

By Theorem 4, the improvement in mean square error, obtained by using the optimal shrinker, over the TSVD baseline, is $3r(\kappa\sigma^2 + (1-\kappa)\tau^2)/\kappa^2$.

## 1.2 Related Work

The general data contamination model we propose includes as special cases several modes extensively studied in the literature, including missing-at-random and outliers. While it is impossible to propose a complete list of algorithms to handle such data, we offer a few pointers, organized around the notions of robust principal component analysis (PCA) and matrix completion. To the best of our knowledge, the precise effect of general data contamination on the SVD (or the closely related PCA) has not been documented thus far. The approach we propose, based on careful manipulation of the data singular values, enjoys three distinct advantages. One, its running time is not prohibitive; indeed, it involves a small yet important modification on top of the SVD or TSVD, so that it is available whenever the SVD is available. Two, it is well understood and its performance (say, in mean square error) can be reliably predicted by the available theory. Three, to the best of our knowledge, none of the approaches below have become mainstream, and most practitioners still turn to the SVD, even in the presence of data contamination. Our approach can easily be used in practice, as it relies on the well-known and very widely used SVD, and can be implemented as a simple modification on top of the existing SVD implementations.

**Robust Principle Component Analysis (RPCA).** In RPCA, one assumes $Y = X + W$ where $X$ is the low rank target matrix and $W$ is a sparse outliers matrix. Classical approaches such as influence functions [20], multivariate trimming [17] and random sampling techniques [14] lack a formal theoretical framework and are not well understood. More modern approaches based on convex optimization [9, 34] proposed reconstructing $X$ from $Y$ via the nuclear norm minimization

$$\min_X ||X||_* + \lambda ||Y - X||_1 ,$$

whose runtime and memory requirements are both prohibitively large in medium and large matrices.

**Matrix Completion.** There are numerous heuristic approaches for data analysis in the presence of missing values [5, 30, 31]. To the best of our knowledge, there are no formal guarantees of their performance. When the target matrix is known to be low rank, the reconstruction problem is known as matrix completion. [7–9] and numerous other authors have shown that a semi-definite program may be used to stably recover the target matrix, even in the presence of additive noise. Here too, the runtime and memory requirements are both prohibitively large in medium and large matrices, making these algorithms infeasible in practice.

## 2 A Unified Model for Uniformly Distributed Contamination

Contamination modes encountered in practice are best described by a combination of primitive modes, shown in Table 1 below. These primitive contamination modes fit into a single template:

**Definition 1.** *Let A and B be two random variables, and assume that all moments of A and B are bounded. Define the* contamination link function

$$f_{A,B}(x) = Ax + B.$$

*Given a matrix X, define the corresponding contaminated matrix Y with entries*

$$Y_{i,j} \overset{indep.}{\sim} f_{A,B}(X_{i,j}). \tag{6}$$

Now observe that each of the primitive modes above corresponds to a different choice of random variables $A$ and $B$, as shown in Table 1. Specifically, each of the primitive modes is described by a different assignment to $A$ and $B$. We employ three different random variables in these assignments: $Z \overset{iid}{\sim} (0, \sigma^2/n)$, a random variable describing multiplicative or additive noise; $W \overset{iid}{\sim} (0, \tau^2/n)$, a random variable describing a large "outlier" measurement; and $M \overset{iid}{\sim} Bernoulli(\kappa)$ describing a random choice of "defective" entries, such as a missing value, an outlier and so on.

Table 1: *Primitive modes fit into the model* (6). *By convention, Y is m-by-n, $Z \overset{iid}{\sim} (0, \sigma^2/n)$ denotes a noise random variable, $W \overset{iid}{\sim} (0, \tau^2/n)$ denotes an outlier random variable and $M \overset{iid}{\sim} Bernoulli(\kappa)$ is a contaminated entry random variable.*

| mode | model | A | B | levels |
|---|---|---|---|---|
| i.i.d additive noise | $Y_{i,j} = X_{i,j} + Z_{i,j}$ | 1 | $Z$ | $\sigma$ |
| i.i.d multiplicative noise | $Y_{i,j} = X_{i,j} Z_{i,j}$ | $Z$ | 0 | $\sigma$ |
| missing-at-random | $Y_{i,j} = M_{i,j} X_{i,j}$ | $M$ | 0 | $\kappa$ |
| outliers-at-random | $Y_{i,j} = X_{i,j} + M_{i,j}W_{i,j}$ | 1 | $MW$ | $\kappa,\tau$ |
| corruption-at-random | $Y_{i,j} = M_{i,j}X_{i,j} + (1 - M_{i,j})W_{i,j}$ | $M$ | $(1-M)W$ | $\kappa,\tau$ |

Actual datasets rarely demonstrate a single primitive contamination mode. To adequately describe contamination observed in practice, one usually needs to combine two or more of the primitive contamination modes into a composite mode. While there is no point in enumerating all possible combinations, Table 2 offers a few notable composite examples, using the framework (6). Many other examples are possible of course.

## 3 Signal Model

Following [32] and [15], as we move toward our formal results we are considering an asymptotic model inspired by Johnstone's Spiked Model [22]. Specifically, we are considering a sequence of increasingly larger data target matrices $X_n$, and corresponding data matrices $Y_n \overset{iid}{\sim} f_{A_n,B_n}(X_n)$. We make the following assumptions regarding the matrix sequence $\{X_n\}$:

**A1** *Limiting aspect ratio:* The matrix dimension $m_n \times n$ sequence converges: $m_n/n \to \beta$ as $n \to \infty$. To simplify the results, we assume $0 < \beta \leq 1$.

**A2** *Fixed signal column span:* Let the rank $r > 0$ be fixed and choose a vector $\mathbf{x} \in \mathbb{R}^r$ with coordinates $\mathbf{x} = (x_1, \ldots x_r)$ such that $x_1 > \ldots > x_r > 0$. Assume that for all $n$

$$X_n = \tilde{U}_n \, diag(x_1, \ldots, x_r)\tilde{V}_n$$

is an arbitrary singular value decomposition of $X_n$,

Table 2: *Some examples of composite contamination modes and how they fit into the model* (6). *Z,W,M are the same as in Table 1.*

| mode | A | B | levels |
|---|---|---|---|
| Additive noise and missing-at-random | $M$ | $ZM$ | $\sigma,\kappa$ |
| Additive noise and corrupt-at-random | $M$ | $ZM + W(1 - M)$ | $\sigma,\kappa,\tau$ |
| multiplicative noise and corrupt-at-random | $ZM$ | $W(1 - M)$ | $\sigma,\kappa,\tau$ |
| Additive noise and outliers | 1 | $Z + W(1 - M)$ | $\sigma,\kappa,\tau$ |

**A3** *Incoherence of the singular vectors of $X_n$:* We make one of the following two assumptions regarding the singular vectors of $X_n$:

**A3.1** $X_n$ is random with an orthogonally invariant distribution. Specifically, $\tilde{U}_n$ and $\tilde{V}_n$, which follow the Haar distribution on orthogonal matrices of size $m_n$ and $n$, respectively.

**A3.2** The singular vectors of $X_n$ are non-concentrated. Specifically, each left singular vector $\tilde{\mathbf{u}}_{n,i}$ of $X_n$ (the $i$-th column of $\tilde{U}_n$) and each right singular vector $\tilde{\mathbf{v}}_{n,j}$ of $X_n$ (the $j$-th column of $\tilde{V}_n$) satisfy[1]

$$||\tilde{\mathbf{u}}_{n,i}||_\infty \leq C\frac{\log^D(m_n)}{\sqrt{m_n}} \qquad \text{and} \qquad ||\tilde{\mathbf{v}}_{n,j}||_\infty \leq C\frac{\log^D(n)}{\sqrt{n}}$$

for any $i, j$ and fixed constants $C, D$.

**Definition 2. (Signal model.)** Let $A_n \overset{\text{iid}}{\sim} (\mu_A, \sigma_A^2/n)$ and $B_n \overset{\text{iid}}{\sim} (0, \sigma_B^2/n)$ have bounded moments. Let $X_n$ follow assumptions **[A1]–[A3]** above. We say that the matrix sequence $Y_n = f_{A_n,B_n}(X_n)$ follows our signal model, where $f_{A,B}(X)$ is as in Definition 1. We further denote $X_n = \sum_{i=1}^{r} x_i\tilde{\mathbf{u}}_{n,i}\tilde{\mathbf{v}}_{n,i}$ for the singular value decomposition of $X_n$ and $Y_n = \sum_{i=1}^{m} y_{n,i}\mathbf{u}_{n,i}\mathbf{v}_{n,i}$ for the singular value decomposition of $Y_n$.

## 4 Main Results

Having described the contamination and the signal model, we can now formulate our main results. All proofs are deferred to the Supporting Information. Let $X_n$ and $Y_n$ follow our signal model, Definition 2, and write $\mathbf{x} = (x_1, \ldots, x_r)$ for the non-zero singular values of $X_n$. For a shrinker $\eta$, we write

$$L_\infty(\eta|\mathbf{x}) \overset{a.s.}{=} \lim_{n\to\infty} \left|\left|\hat{X}_n(Y_n) - X_n\right|\right|_F^2.$$

assuming the limit exists almost surely. The special case of hard thresholding at $\lambda$ is denoted as $L_\infty(\eta|\mathbf{x})$.

**Definition 3. Optimal shrinker and optimal threshold.** A shrinker $\eta^*$ is called *optimal* if

$$L_\infty(\eta|\mathbf{x}) \leq L_\infty(\eta|\mathbf{x})$$

for any shrinker $\eta$, any $r \geq 1$ and any $\mathbf{x} = (x_1, \ldots, x_r)$. Similarly, a threshold $\lambda$ is called optimal if $L_\infty(\lambda^*|\mathbf{x}) \leq L_\infty(\lambda|\mathbf{x})$ for any threshold $\lambda$, any $r \geq 1$ and any $\mathbf{x} = (x_1, \ldots, x_r)$.

With these definitions, our main results Theorem 2 and Theorem 1 become formal. To make Theorem 3 formal, we need the following lemma and definition.

**Lemma 1. Decomposition of the asymptotic mean square error.** Let $X_n$ and $Y_n$ follow our signal model (Definition 2) and write $\mathbf{x} = (x_1, \ldots, x_r)$ for the non-zero singular values of $X_n$, and let $\eta$ be the optimal shrinker. Then the limit $L_\infty(\eta|\mathbf{x})$ a.s. exists, and $L_\infty(\eta|\mathbf{x}) \overset{a.s.}{=} \sum_{i=1}^{r} L_1(\eta|x)$, where

$$L_1(\eta|x) = \begin{cases} x^2\left(1 - \dfrac{(t^4 - \beta)^2}{(t^4 + \beta t^2)(t^4 + t^2)}\right) & t \geq \beta^{\frac{1}{4}} \\ x^2 & t < \beta^{\frac{1}{4}} \end{cases}$$

where $t = (\mu_A \cdot x)/\sigma_B$. Similarly, for a threshold $\lambda$ we have $L_\infty(\lambda|\mathbf{x}) = \sum_{i=1}^{r} L_1(\lambda|x)$ with

$$L_1(\lambda|x) = \begin{cases} \left(\dfrac{\sigma_B}{\mu_A}\right)^2\left(\left(t + \dfrac{1}{t}\right)\left(t + \dfrac{\beta}{t}\right) - \left(t^2 - \dfrac{2\beta}{t^2}\right)\right) & \mu_A x \geq x(\lambda) \\ x^2 & \mu_A x < x(\lambda) \end{cases}$$

Where

$$x(y) = \begin{cases} (\sigma_B/\sqrt{2}\mu_A)\sqrt{(y/\sigma_B)^2 - \beta - 1 + \sqrt{\left(1 + \beta - (y/\sigma_B)^2\right)^2 - 4\beta}} & t \geq \beta^{\frac{1}{4}} \\ 0 & t < \beta^{\frac{1}{4}} \end{cases} \tag{7}$$

**Definition 4.** Let $\eta_0$ be the zero singular value shrinker, $\eta_0(y) \equiv 0$, so that $\hat{X}_{\eta_0}(Y) \equiv 0$. Let $\eta$ be a singular value shrinker. The critical signal level for $\eta$ is

$$x^{critical}(\eta) = \inf_x \{ L_1(\eta|X) < L_1(\eta_0|X) \}$$

As we can see, the asymptotic mean square error decomposes over the singular values of the target matrix, $x_1, \ldots, x_r$. Each value $x_i$ that falls below $x^{critical}(\eta)$ is better estimated with the zero shrinker $\eta_0$ than with $\eta$. It follows that any $x_i$ that falls below $x^{critical}(\eta^*)$, where $\eta^*$ is the optimal shrinker, cannot be reliably estimated by any shrinker $\eta$, and its corresponding data singular value $y_i$ should simply be set to zero. This makes Theorem 2 formal.

## 5 Estimating the model parameters

In practice, using the optimal shrinker we propose requires an estimate of the model parameters. In general, $\sigma_B$ is easy to estimate from the data via a median-matching method [15], namely

$$\hat{\sigma}_B = \frac{y_{med}}{\sqrt{n \mu_\beta}},$$

where $y_{med}$ is the median singular value of Y, and $\mu_\beta$ is the median of the Marčenko-Pastur distribution. However, estimation of $\mu_A$ and $\sigma_A$ must be considered on a case-by-case basis. For example, in the "Additive noise and missing at random" mode (table 2), $\sigma_A \equiv 1$ is known, and $\mu_A$ is estimated by dividing the amount of missing values by the matrix size.

## 6 Simulation

Simulations were performed to verify the correctness of our main results[2]. For more details, see Supporting Information.

1. **Critical signal level $x^{critical}(\lambda^*)$ under increasing noise.** Figure 2, left panel, shows the amount of data singular values $y_i$ above $x^{critical}(\lambda^*)$, as a function of the fraction of missing values $\kappa$. Theorem 3 correctly predicts the exact values of $\kappa$ at which the "next" data singular value falls below $x^{critical}(\lambda^*)$.

2. **Phase plane for critical signal levels $x^{critical}(\eta^*)$ and $x^{critical}(\lambda^*)$.** Figure 1, right panel, shows the $x, \kappa$ plane, where $x$ is the signal level and $\kappa$ is the fraction of missing values. At each point in the plane, several independent data matrices were generated. Heatmap shows the fraction of the experiments at which the data singular value $y_1$ was above $x^{critical}(\eta^*)$ and $x^{critical}(\lambda^*)$. The overlaid graphs are theoretical predictions of the critical points.

3. **Brute-force verification of the optimal shrinker shape.** Figure 2, right panel, shows the shape of the optimal shrinker (Theorem 1). We performed a brute-force search for the value of $\eta(y)$ that produces the minimal mean square error. A brute force search, performed with a relatively small matrix size, matches the asymptotic shape of the optimal shrinker.

## 7 Conclusions

Singular value shrinkage emerges as an effective method to reconstruct low-rank matrices from contaminated data that is both practical and well understood. Through simple, carefully designed manipulation of the data singular values, we obtain an appealing improvement in the reconstruction mean square error. While beyond our present scope, following [16], it is highly likely that the optimal shrinker we have developed offers the same mean square error, asymptotically, as the best rotation-invariant estimator based on the data, making it asymptotically the best SVD-based estimator for the target matrix.

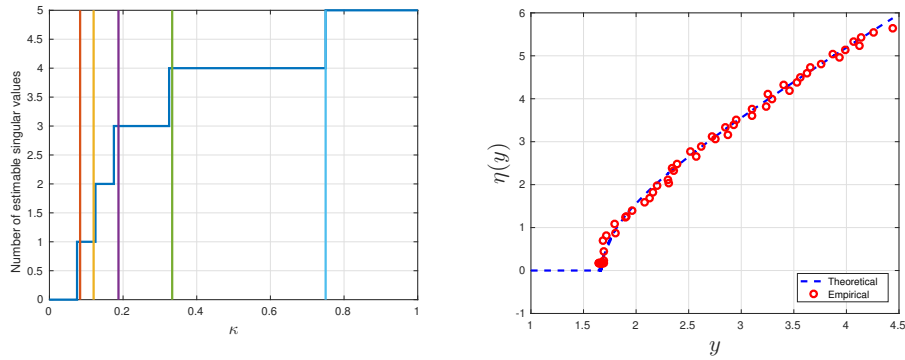

Figure 2: Left: empirical validation of the predicted critical signal level (Simulation 1). Right: Empirical validation of the optimal shrinker shape (Simulation 3).

## Acknowledgements

DB was supported by Israeli Science Foundation grant no. 1523/16 and German-Israeli Foundation for scientific research and development program no. I-1100-407.1-2015.

## Footnotes

[1]The incoherence assumption is widely used in related literature [6, 12, 27], and asserts that the singular vectors are spread out so $X$ is not sparse and does not share singular subspaces with the noise.

[2]The full Matlab code that generated the figures in this paper and in the Supporting Information is permanently available at `https://purl.stanford.edu/kp113fq0838`.

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
