[Supplementary Material · nips_2017_SI.pdf]

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

.** Consider the composite mode "i.i.d additive noise and missing-at-random"(sec. 2, mode 2) with noise level $\sigma = 1$, singular values $\mathbf{x} = (2, 3, 4, 5, 6)$ and matrix size $m = n = 1000$. To test whether our main results correctly predict the values of $\kappa$ at which the signal singular values stop being estimable, we scanned the $\kappa$ axis. At each value of $\kappa$ we counted the number of estimable singular values and compared with the cutoff values of $\kappa$, at which this number should change. Figure 2 shows the number of measured estimable singular values against contamination level $\kappa$, with overlaid cut levels predicted by our main results (vertical lines).

Figure 2: Left: empirical validation of the predicted critical signal level (Simulation 1). Right: Empirical validation of the optimal shrinker shape (Simulation 3).

2. **Phase plane for critical signal levels** $x^{critical}(\eta^*)$ **and** $x^{critical}(\lambda^*)$**.** Figure 1, right panel, shows the $x, \kappa$ plane, where $x$ is the signal level and $\kappa$ is the fraction of missing values. At each point in the plane, several independent experiments (data matrices) were generated. The heatmap shows the the fraction of the experiments at which the data singular value $y_1$ was above $x^{critical}(\eta^*)$ and $x^{critical}(\lambda^*)$. Overlaid are the theoretical prediction of the critical points. Matrix size is $600 \times 600$, Monte carlo is 5 and $\beta = 1$.

3. **Brute-force verification of the optimal shrinker shape.** Figure (2,right) shows the shape of the optimal shrinker (Theorem 1). We performed a brute-force calculation scanning for the value of $\eta(y)$ that produces the lower mean square error. Brute force search, peformed in a $250 \times 250$ matrix, matches the asymptotic shape of the optimal shrinker. The scan set the noise level $\sigma$ to 1, missing data level to $1 - \mu_A = 0.3$, $\beta = m/n = 1$, and scanned signal values in range [0,6].

# 8   Proofs

In this section we prove our main results: Theorem 1, Theorem 2, Theorem 3 and Theorem 4, as well as Lemma 1. The proofs rely on the following fundamental lemmas.

**Lemma 2.** Let $Y_n = f_{A_n, B_n}(X_n)$ be a matrix sequence following our signal model (Definition 2) with $A_n \overset{\text{iid}}{\sim} (\mu_A, \sigma_A^2/n)$ and $B_n \overset{\text{iid}}{\sim} (0, \sigma_B^2/n)$. Write $\bar{x}_i = \mu_A \cdot x_i$. Then the following holds:

1. For each $1 \leq i \leq r$ we have

$$
\lim_{n \to \infty} y_{n,i} \overset{a.s.}{=} \begin{cases} \sigma_B \sqrt{\left( \dfrac{\bar{x}_i}{\sigma_B} + \dfrac{\sigma_B}{\bar{x}_i} \right) \left( \dfrac{\bar{x}_i}{\sigma_B} + \dfrac{\beta \sigma_B}{\bar{x}_i} \right)} & \bar{x}_i > \sigma_B \beta^{\frac{1}{4}} \\ \sigma_B (1 + \sqrt{\beta}) & \bar{x}_i \leq \sigma_B \beta^{\frac{1}{4}} \end{cases}
$$

2. Let $1 \leq i \leq r$ and $1 \leq j \leq m_n$. If $\bar{x}_i > \sigma_B \beta^{\frac{1}{4}}$ and $j \leq r$, we have

$$
d \cdot \lim_{n \to \infty} |\langle \tilde{\mathbf{u}}_i, \mathbf{u}_{n,j} \rangle|^2 \overset{a.s.}{=} \begin{cases} \dfrac{(\frac{\bar{x}_i}{\sigma})^4 - \beta}{(\frac{\bar{x}_i}{\sigma})^4 + \beta(\frac{\bar{x}_i}{\sigma})^2} & \bar{x}_i = \bar{x}_j \\ 0 & \bar{x}_i \neq \bar{x}_j \end{cases} \tag{8}
$$

and

$$
d \cdot \lim_{n \to \infty} |\langle \tilde{\mathbf{v}}_i, \mathbf{v}_{n,j} \rangle|^2 \overset{a.s.}{=} \begin{cases} \dfrac{(\frac{\bar{x}_i}{\sigma})^4 - \beta}{(\frac{\bar{x}_i}{\sigma})^4 + (\frac{\bar{x}_i}{\sigma})^2} & \bar{x}_i = \bar{x}_j \\ 0 & \bar{x}_i \neq \bar{x}_j \end{cases}
$$

Where $d_i = |\{x_j | x_j = x_i\}|$. Otherwise, both quantities converge a.s to 0 as $n \to \infty$.

**Proof of Theorem 1.** By lemma 3 we have that for any singular value hard threshold estimator $\lambda$, AMSE is given by (11). As for the bottom case in (11), clearly $\overline{x}^2$ is strictly increasing in $\overline{x}$, and therefor in $x$ as well. As for the top case in (11), direct differentiation shows that it is strictly decreasing. It follows that the two functions of $\overline{x}$ intersect at a unique point. Denote their intersection point by $\overline{x^{est}} = \mu_A x^{est}$:

$$\overline{x^{est}} = \sigma_B \sqrt{\frac{1 + \beta + \sqrt{1 + 14\beta + \beta^2}}{2}}.$$

From the displacement formula 18, it follows that the following hard threshold for Y's singular values achieves the minimum AMSE between the above two expressions:

$$\lambda^* = \sigma_B \left( \sqrt{ \left( \frac{\overline{x^{est}}}{\sigma_B} + \frac{\sigma_B}{\overline{x^{est}}} \right) \left( \frac{\overline{x^{est}}}{\sigma_B} + \frac{\beta \sigma_B}{\overline{x^{est}}} \right) } \right)$$

. $\qquad\qquad\qquad\qquad\qquad\qquad\qquad\qquad\qquad\qquad\qquad\qquad\qquad$ $\square$

**Proof of Theorem 2.** Recall that for any matrix $X$, $||X||_F^2 = \langle X, X \rangle$ where $\langle \cdot, \cdot \rangle$ is the Hilbert–Schmidt matrix inner product. The notation $\langle X, X \rangle$ refers to Hilbert–Schmidt matrix inner product or to the standard Euclidean inner product. Denoting our shrinker by $\eta = \eta(y)$, and let $\overline{\eta} = \frac{\eta(y_i)}{\mu_A}$, we evaluate the mean square error:

$$\left\| \frac{1}{\mu_A} \hat{X}_\eta(Y) - X \right\|_F^2 = \langle \frac{1}{\mu_A} \hat{X}_\eta(Y) - X, \frac{1}{\mu_A} \hat{X}_\eta(Y) - X \rangle_{HS} =$$

$$\langle \frac{1}{\mu_A} \hat{X}_\eta(Y), \frac{1}{\mu_A} \hat{X}_\eta(Y) \rangle + \langle X_n, X_n \rangle - 2 \langle \frac{1}{\mu_A} \hat{X}_\eta(Y_n), X_n \rangle =$$

$$\sum_{i=1}^{m_n} \overline{\eta}(y_{n,i})^2 + \sum_{i=1}^{r} x_i^2 - 2 \sum_{i,j=1}^{r} x_i \overline{\eta}(y_{n,i}) \langle \mathbf{a_i} \mathbf{b_i'}, \mathbf{u_{n,j}} \mathbf{v_{n,j}'} \rangle =$$

$$\sum_{i=r+1}^{m_n} \overline{\eta}(y_{n,i})^2 + \sum_{i=1}^{r} \left[ \overline{\eta}(y_{n,i})^2 + x_i^2 - 2 x_i \sum_{j=1}^{r} \overline{\eta}(y_{n,j}) \langle \mathbf{a_i} \mathbf{b_i'}, \mathbf{u_{n,j}} \mathbf{v_{n,j}'} \rangle \right].$$

We now assume that $\forall i, y_i < \sigma(1 + \sqrt{\beta}), \overline{\eta}(y_i) = 0$; this is justified inside the proof of 3. As the first step of our analysis, we also assume that $rank(X) = 1$ to get when $n \to \infty$:

$$\left\| \frac{1}{\mu_A} \hat{X}_\eta(Y) - X \right\|_F^2 \stackrel{a.s.}{=} \overline{\eta}(y_{n,1})^2 + x_1^2 - 2 x_1 \overline{\eta}(y_{n,1}) \langle \mathbf{a_1} \mathbf{b_1'}, \mathbf{u_{n,1}} \mathbf{v_{n,1}'} \rangle =$$

$$\overline{\eta}(y_{n,1})^2 + x_1^2 - 2 x_1 \overline{\eta}(y_{n,1}) \langle \mathbf{a_1}, \mathbf{u_{n,1}} \rangle \langle \mathbf{b_1'}, \mathbf{v_{n,1}'} \rangle.$$

Differentiating w.r.t $\eta$ and comparing to zero we get: $\overline{\eta} = x_1 \langle \mathbf{a_1}, \mathbf{u_{n,1}} \rangle \langle \mathbf{b_1'}, \mathbf{v_{n,1}'} \rangle$. We would like to express $\overline{\eta}$ as a function of $y = y_1$ instead of $x_1$. When $n \to \infty$, the expression $\langle \mathbf{a_1}, \mathbf{u_{n,1}} \rangle \langle \mathbf{b_1'}, \mathbf{v_{n,1}'} \rangle$ is known. $x_1$ can be expressed as a function of $y$ as in 7. Denote $t = \frac{\mu_A \cdot x}{\sigma_B}$, then we have:

$$\overline{\eta} = x_1 \langle \mathbf{a_1}, \mathbf{u_{n,1}} \rangle \langle \mathbf{b_1'}, \mathbf{v_{n,1}'} \rangle = \sigma_B \cdot \frac{t^4 - \beta}{\sqrt{(t^4 + \beta t^2)(t^4 + t^2)}} = \frac{\sigma_B^2}{y} \frac{t^4 - \beta}{t^2} = \frac{\sigma_B^2}{y} \left( \sqrt{t^2 - \frac{\beta}{t^2}} \right)^2$$

$$= \frac{\sigma_B^2}{y} \sqrt{(t^2 + \frac{\beta}{t^2})^2 - 4\beta} = \frac{\sigma_B^2}{y} \sqrt{((t + \frac{1}{t})(t + \frac{\beta}{t}) - \beta - 1))^2 - 4\beta} = \frac{\sigma_B^2}{y} \sqrt{(\frac{y}{\sigma_B}^2 - \beta - 1)^2 - 4\beta}.$$

Finally:

$$\eta^* = \begin{cases} \dfrac{\sigma_B^2}{\mu_A y} \sqrt{((\frac{y}{\sigma_B})^2 - \beta - 1)^2 - 4\beta} & y \geq \sigma_B(1 + \sqrt{\beta}) \\ 0 & y < \sigma_B(1 + \sqrt{\beta}) \end{cases}$$

Now without the assumption that $rank(X) = 1$, we have:

$$\left|\left|\hat{X}_\eta(Y) - X\right|\right|_F \overset{a.s.}{=} \sum_{i=i}^{r} \eta(y_{n,i})^2 + x_i^2 - 2x_i\eta(y_{n,i})\langle \mathbf{a_i}, \mathbf{u_{n,i}}\rangle\langle \mathbf{b'_i}, \mathbf{v'_{n,i}}\rangle.$$

Here there are $r$ independent positive summands; the minimum of each is achieved by $\eta^*$ above 8. □

**Proof of Theorem 3.** We first calculate $x^{critical}(\lambda^*)$.

Let $\overline{x} = \mu_A x$, and $\lambda$ a threshold operator s.t. $\lambda \geq \sigma_B(1 + \sqrt{\beta})$. Denote by $\overline{x^{est}} = \mu_A x^{est}$ be the unique solution to the intersection equality of (3):

$$\overline{x}^2 = \sigma_B^2((\frac{\overline{x}}{\sigma_B} + \frac{\sigma_B}{\overline{x}})(\frac{\overline{x}}{\sigma_B} + \frac{\beta\sigma_B}{\overline{x}}) - ((\frac{\overline{x}}{\sigma_B})^2 - \frac{2\beta\sigma_B^2}{\overline{x}^2}))$$

The left and right hand expressions denote the zero estimator loss $L_1(\lambda_0|X)$ and $L(\lambda|X)$ for a given $\lambda$ accordingly. For $x < x^{est} = \frac{\sigma_B}{\mu_A}c$, $L_1(\lambda_0|X) < L(\lambda|X)$ for any $\lambda$, and the proof for $x^{critical}(\lambda^*)$ follows.

We now turn to $x^{critical}(\eta^*)$. According to the fundamental displacement lemma (lemma 2 ), $x_i$ is asymptotically undetectable– displaced to a value independent of x – if its matching limit data singular value $y_i \leq \sigma_B(1 + \sqrt{\beta})$. The transition between detectable and undetectable is at:

$$\overline{x_i} = \sigma_B\beta^{\frac{1}{4}}$$

Assume a shrinker $\eta$ different from the zero shrinker, s.t. for $y_i \leq \sigma_B(1 + \sqrt{\beta})$, $\eta(y_i) \neq 0$. The AMSE expression for any shrinker is:

$$\left|\left|\frac{1}{\mu_A}\hat{X}_\eta(Y) - X\right|\right|_F^2 = \langle \frac{1}{\mu_A}\hat{X}_\eta(Y) - X, \frac{1}{\mu_A}\hat{X}_\eta(Y) - X\rangle_{HS} =$$

$$\langle \frac{1}{\mu_A}\hat{X}_\eta(Y), \frac{1}{\mu_A}\hat{X}_\eta(Y)\rangle + \langle X_n, X_n\rangle - 2\langle \frac{1}{\mu_A}\hat{X}_\eta(Y_n), X_n\rangle =$$

$$\sum_{i=1}^{m_n} \overline{\eta}(y_{n,i})^2 + \sum_{i=1}^{r} x_i^2 - 2\sum_{i,j=1}^{r} x_i\overline{\eta}(y_{n,i})\langle \mathbf{a_i b'_i}, \mathbf{u_{n,j} v'_{n,j}}\rangle$$

For $\eta$ as assumed above, the expression $\sum_{i=1}^{m_n} \overline{\eta}(y_{n,i})^2$ tends to $\infty$ with $n$: there are infinitely many noise singular values that are shrunk to a positive value. The other summands are bounded, so $L(\eta|X) < L(\eta_0|X)$ and therefore,

$$x^{critical}(\eta^*) = \frac{\sigma_B\beta^{\frac{1}{4}}}{\mu_A}.$$

□

**Proof of Theorem 4.** We start with $M_{\lambda^*}$. From lemma 3 the maximal AMSE is at the intersection point of the AMSE expressions. This point is given by:

$$\overline{x^{est}} = \mu_A x^{est} = \sigma_B\sqrt{\frac{1 + \beta + \sqrt{1 + 14\beta + \beta^2}}{2}}$$

Given $\beta = 1$, plug in $x^{est}$ and the calculation of $M_{\lambda^*}$ follows.

Turning to $M_{TSVD}$, TSVD is actually hard thresholding at the bulk edge. According to the displacement formula 1, $\lambda_{TSVD} = \sigma_B(1 + \sqrt{(\beta)})$, thus signal slightly larger than $x^{worst} = \sigma_B\beta^{\frac{1}{4}}/\mu_A$, will displace into a matching $y$ value at slightly larger size than the bulk edge, and so the top loss term in

(11) applies. Direct differentiation shows that it is strictly decreasing. Noting that the first term is larger than $x^2$,the proof follows.

Turning finally to $M_{\eta^*}$, similarly to when we derived the optimal shrinker, we begin by focusing on a single singular value. Considering the optimal shrinker for a rank one matrix:

$$\eta = x_1 \langle \mathbf{a_1}, \mathbf{u_{n,1}} \rangle \langle \mathbf{b'_1}, \mathbf{v'_{n,1}} \rangle,$$

and let $t = (\mu_A \cdot x)/\sigma_B$, the AMSE is:

$$\left\| \hat{X}_\eta(Y) - \mu_A X \right\|_F^2 = \eta(y_{n,1})^2 + \overline{x_1}^2 - 2\overline{x_1}\eta(y_{n,1})\langle \mathbf{a_1}, \mathbf{u_{n,1}} \rangle \langle \mathbf{b'_1}, \mathbf{v'_{n,1}} \rangle \overset{a.s.}{=}$$

$$\overline{x}^2 - \eta^2 =$$

$$\mu^2 x^2 \left( 1 - \frac{(t^4 - \beta)^2}{(t^4 + \beta t^2)(t^4 + t^2)} \right)$$

Choosing $t^2 = z$, we get

$$\sigma^2 z \left( 1 - \frac{(z^2 - \beta)^2}{(z^2 + \beta z)(z^2 + z)} \right)$$

According to the assumption on $x$, $t > \beta^{\frac{1}{4}}$ and therefore $z > \sqrt{\beta}$ and the expression is monotonically increasing with $\beta$, in $\beta \in (0, 1]$. Setting $\beta = 1$, the simplified expression is

$$\sigma^2 \left( 2 - \frac{1}{z} \right) \underset{z \to \infty}{\to} 2\sigma^2$$

This is the expression for the squared asymptotic loss for every singular value, yielding worst case AMSE of $r2\sigma^2$ in total. Since the loss is measured against $X$ and not $\mu_A X$, we multiply the expressions by a factor of $\frac{1}{\mu_A^2}$ and the proof follows.

$\square$

**Proof of Lemma 1.** We begin with the shrinker loss.

$$L_\infty(\eta|\mathbf{x}) = \left\| \frac{1}{\mu_A}\hat{X}_\eta(Y) - X \right\|_F^2 = \langle \frac{1}{\mu_A}\hat{X}_\eta(Y) - X, \frac{1}{\mu_A}\hat{X}_\eta(Y) - X \rangle_{HS}$$

$$= \langle \frac{1}{\mu_A}\hat{X}_\eta(Y), \frac{1}{\mu_A}\hat{X}_\eta(Y) \rangle + \langle X_n, X_n \rangle - 2\langle \frac{1}{\mu_A}\hat{X}_\eta(Y_n), X_n \rangle$$

$$= \sum_{i=1}^{m_n} \overline{\eta}(y_{n,i})^2 + \sum_{i=1}^{r} x_i^2 - 2\sum_{i,j=1}^{r} x_i\overline{\eta}(y_{n,i})\langle \mathbf{a_i b'_i}, \mathbf{u_{n,j} v'_{n,j}} \rangle$$

$$= \sum_{i=r+1}^{m_n} \overline{\eta}(y_{n,i})^2 + \sum_{i=1}^{r} \left[ \overline{\eta}(y_{n,i})^2 + x_i^2 - 2x_i\sum_{j=1}^{r} \overline{\eta}(y_{n,j})\langle \mathbf{a_i b'_i}, \mathbf{u_{n,j} v'_{n,j}} \rangle \right].$$

Assuming that $\forall y_i < \sigma_B(1 + \sqrt{B}), \overline{\eta}(y_i) = 0$,when $n \to \infty$:

$$\overset{a.s.}{=} \sum_{i=i}^{r} \eta(y_{n,i})^2 + x_i^2 - 2x_i\eta(y_{n,i})\langle \mathbf{a_i}, \mathbf{u_{n,i}} \rangle \langle \mathbf{b'_i}, \mathbf{v'_{n,i}} \rangle$$

The AMSE decomposition for a hard threshold $\lambda$ is a result of lemma 3.

$\square$

**Lemma 3. AMSE of hard threshold $\lambda$.** Fix the signal matrix's rank $r$ and let $\underline{x} \in \mathbb{R}^r$ the fixed rank singular values vector of the signal $X$. Let $\{X_n(\underline{x})\}_{n=1}^\infty$, $\{A_n\}_{n=1}^\infty$ and $\{B_n\}_{n=1}^\infty$, Let $\beta$ be the sequence asymptotic ratio $X_n \in \mathbb{R}^{m_n \times n}(\mathbb{R})$ where $\lim_{n \to \infty} m_n/n = \beta$. Let $Y_n$ matrix

sequence s.t. $Y = A \odot X + B$ as in the basic framework (2) with matching $\beta, \sigma_B, \mu_A$. Let $Y$'s SVD decomposition be:

$$Y = \sum_{i=1}^{m} y_i \mathbf{u}_i \mathbf{v}_i' \tag{9}$$

Assume we estimate X by the hard threshold estimator with parameter k, i.e. by setting to zero $Y$'s singular values that are smaller than $k$:

$$\hat{X}_\lambda(Y) = \sum_{i=1}^{m} \eta_H(y_i; \lambda) \mathbf{u}_i \mathbf{v}_i', \text{, where } \eta_H(y; \lambda) = \begin{cases} 0 & y < \lambda, \\ y & \text{otherwise} \end{cases}$$

and let $\lambda$ be a selected hard threshold s.t. $\lambda \geq \sigma_B(1 + \sqrt{\beta})$.

Define the Asymptotic MSE:

$$AMSE(\hat{X}_\lambda(Y)) = \lim_{n \to \infty} \left\| \frac{1}{\mu_A} \hat{X}_\lambda(Y_n) - X_n \right\|_F^2$$

and . Then

$$AMSE(\hat{X}_\lambda, \mathbf{x}) = \sum_{i=1}^{r} M(\hat{X}_\lambda, x_i) \tag{10}$$

and

$$M(\hat{X}_\lambda, x) = \begin{cases} \left(\frac{\sigma_B}{\mu_A}\right)^2 \left(\left(t + \frac{1}{t}\right)\left(t + \frac{\beta}{t}\right) - \left(t^2 - \frac{2\beta}{t^2}\right)\right) & \overline{x} \geq x(\lambda) \\ x^2 & \overline{x} < x(\lambda) \end{cases} \tag{11}$$

where $t = \mu_A x / \sigma_B$.

$$x(y) = \begin{cases} (\sigma_B/\sqrt{2}\mu_A)\sqrt{(y/\sigma_B)^2 - \beta - 1 + \sqrt{(1 + \beta - (y/\sigma_B)^2)^2 - 4\beta}} & y \geq \sigma_B\sqrt{1 + \beta + \sqrt{\beta}(1 + \mu_A^2)} \\ 0 & else \end{cases} \tag{12}$$

**Proof of lemma 3**  We'll denote the SVD of a signal matrix X, which is an element of $\{X_n\}$, by:

$$X = \sum_{i=1}^{r} x_i \mathbf{a}_i \mathbf{b}_i'$$

$$\left\| \hat{X}_\lambda(Y) - \mu_A X \right\|_F^2 = \langle \hat{X}_\lambda(Y) - \mu_A X, \hat{X}_\lambda(Y) - \mu_A X \rangle_{HS} =$$

$$\langle \hat{X}_\lambda(Y), \hat{X}_\lambda(Y) \rangle + \langle \mu_A X_n, \mu_A X_n \rangle - 2\langle \hat{X}_\lambda(Y_n), \mu_A X_n \rangle =$$

$$\sum_{i=1}^{m_n} \eta_H(y_{n,i}; \lambda)^2 + \sum_{i=1}^{r} \overline{x_i}^2 - 2 \sum_{i,j=1}^{r} \overline{x_i} \eta_H(y_{n,j}; \lambda) \langle \mathbf{a}_i \mathbf{b}_i', \mathbf{u}_{n,j} \mathbf{v}_{n,j}' \rangle =$$

$$\sum_{i=r+1}^{m_n} \mu_H(y_{n,i}; \lambda)^2 + \sum_{i=1}^{r} \left[ (\mu_H(y_{n,i}; \lambda)^2 + \overline{x_i}^2) - 2\overline{x_i} \sum_{j=1}^{r} \mu_H(y_{n,j}; \lambda) \langle \mathbf{a_i b_i'}, \mathbf{u_{n,j} v_{n,j}'} \rangle \right].$$

According to the displacement formula 1

1. $y_{n,r+1} \overset{a.s.}{=} \sigma_B(1 + \sqrt{\beta}) < \lambda$, the leftmost term above converges almost surely to zero.

2. When $0 \leq \overline{x_i} \leq \sigma_B \beta^{\frac{1}{4}}$, all that remains is $\sum_{i=1}^{r} \overline{x_i}^2$ which proves the second case of lemma 3.

Assuming now $\overline{x_i} \geq \sigma_B \beta^{\frac{1}{4}}$, we calculate the limit of every expression in:

$$\sum_{i=1}^{r} \left[ (\mu_H(y_{n,i}; \lambda)^2 + \overline{x_i}^2) - 2\overline{x_i} \sum_{j=1}^{r} \mu_H(y_{n,j}; \lambda) \langle \mathbf{a_i b_i'}, \mathbf{u_{n,j} v_{n,j}'} \rangle \right].$$

1. According to the displacement formula:

$$\lim_{n \to \infty} \eta_H(y_{n,i}; \lambda)^2 \overset{a.s.}{=} \begin{cases} \sigma_B^2 (\frac{\overline{x_i}}{\sigma_B} + \frac{\sigma_B}{\overline{x_i}})(\frac{\overline{x_i}}{\sigma_B} + \frac{\beta \sigma_B}{\overline{x_i}}) & \text{for } y_{n_i} > \lambda \text{ ;} \\ 0 & \text{otherwise} \end{cases}$$

2. According to the rotation formula:

$$\lim_{n \to \infty} \langle \mathbf{a_i b_i'}, \mathbf{u_{n,j} v_{n,j}'} \rangle = \lim_{n \to \infty} \langle \mathbf{a_j}, \mathbf{u_{n,j}} \rangle \langle \mathbf{b_i}, \mathbf{v_{n,j}} \rangle \overset{a.s.}{=} \begin{cases} \dfrac{(\frac{\overline{x_i}}{\sigma_B})^4 - \beta}{d_i \sqrt{((\frac{\overline{x_i}}{\sigma_B})^4 + \beta(\frac{\overline{x_i}}{\sigma_B})^2)((\frac{\overline{x_i}}{\sigma_B})^4 + (\frac{\overline{x_i}}{\sigma_B})^2)}} & \text{for } \overline{x_i} = \overline{x_j}; \\ 0 & \text{otherwise} \end{cases}$$

   where $d_i = |\{x_j | x_j = \overline{x_i}\}|$.

3. 

$$\lim_{n \to \infty} \mu_H(y_{n,j}; \lambda) \langle \mathbf{a_i b_i'}, \mathbf{u_{n,j} v_{n,j}'} \rangle \overset{a.s.}{=} \begin{cases} \dfrac{\sigma_B ((\frac{\overline{x_i}}{\sigma_B})^4 - \beta)}{(\frac{\overline{x_i}}{\sigma_B})^3} & \text{for } y_{n_i} > \lambda; \\ 0 & \text{otherwise} \end{cases}$$

4. For the last and rightmost element of the AMSE, assuming $\forall i, d_i = 1$:

$$-2\overline{x_i} \mu_H(y_{n,j}; \lambda) \langle \mathbf{a_i}, \mathbf{v_{n,j}} \rangle \langle \mathbf{b_i}, \mathbf{u_{n,j}} \rangle \overset{a.s.}{=} -\sigma_B^2 \left( \frac{2x_i^2}{\sigma_B^2} - 2\beta \frac{\sigma^2}{x_i^2} \right)$$

Adding all expressions yields for $\overline{x_i} \geq \sigma_B \beta^{\frac{1}{4}}$ and $\overline{x_i} > x(\lambda)$:

$$M(\hat{X}_\lambda, x) = \sigma_B^2 ((\frac{\overline{x_i}}{\sigma_B} + \frac{\sigma_B}{\overline{x_i}})(\frac{\overline{x_i}}{\sigma_B} + \frac{\beta \sigma_B}{\overline{x_i}}) - ((\frac{\overline{x_i}}{\sigma_B})^2 - \frac{2\beta \sigma_B^2}{\overline{x_i}^2}))$$

. Which concludes the calculation by showing the first expression in the lemma. Since the loss is measured against $X$ and not $\mu_A X$, we multiply the expressions by a factor of $\frac{1}{\mu_A^2}$ and the proof follows.

$\square$

It remains to prove Lemma 2. The proof follows a strategy proposed by [27]. It is convenient to break the proof into a sequence of lemmas, which are of independent interest.

**Definition 5.** Let $Z$ be an $m$-by-$n$ matrix. For $\mathbf{u}_1, \mathbf{u}_2 \in \mathbb{R}^m$ and $\mathbf{v}_1, \mathbf{v}_2 \in \mathbb{R}^n$, define

$$H(w | \mathbf{u}_1, \mathbf{u}_2, Z) := \mathbf{u}_1'(w^2 I_n - ZZ')^{-1} \mathbf{u}_2 \tag{13}$$

$$Q(w | \mathbf{v}_1, \mathbf{v}_2, Z) := \mathbf{v}_1'(w^2 I_n - Z'Z)^{-1} \mathbf{v}_2. \tag{14}$$

**Definition 6.** Let $\mathcal{Z} = \{Z_n\}_{n=0}^{\infty}$ be a sequence of matrices s.t. $Z_n$ is $m_n$-by-$n$ and $\lim_{n \to \infty} m_n/n = \beta$. Let $\mathcal{U} = \{U_n\}_{n=0}^{\infty}$ be any sequence of $m_n$-by-$m_n$ orthonormal matrices with columns $U_n = (\mathbf{u}_{n,1}, \ldots, \mathbf{u}_{n,m_n})$. Similarly let $\mathcal{V} = \{V_n\}_{n=0}^{\infty}$ be any sequence of $n$-by-$n$ orthonormal matrices with columns $V_n = (\mathbf{v}_{n,1}, \ldots, \mathbf{v}_{n,n})$. Define

$$H_{i,j}(w | \mathcal{U}, \mathcal{Z}) := \lim_{n \to \infty} H(w | \mathbf{u}_{n,i}, \mathbf{u}_{n,j}, Z_n) \tag{15}$$

$$Q_{i,j}(w | \mathcal{V}, \mathcal{Z}) := \lim_{n \to \infty} H(w | \mathbf{v}_{n,i}, \mathbf{v}_{n,j}, Z_n) \tag{16}$$

assuming these limits exist almost surely.

The following result is due to Bloemendal et al [2]:

**Lemma 4.** Let $\mathcal{Z} = \{Z_n\}_{n=0}^{\infty}$ a sequence of matrices s.t. $Z_n$ is $m_n$-by-$n$ and $\lim_{n\to\infty} m_n/n = \beta \leq 1$. Further assume that $(Z_n)_{i,j} \overset{iid}{\sim} \mathcal{F}$, where $\mathcal{F}$ is some distribution with bounded moments, mean 0 and variance $\sigma^2/n$. Let $\mathcal{U}$ and $\mathcal{V}$ be arbitrary sequences of orthonormal matrices as in Definition 6, which are either nonrandom or independent of $\{Z_n\}$. Then for all $1 \leq i \leq m_n$ and $1 \leq j \leq n$,

$$H_{i,j}(w \mid \mathcal{U}, \mathcal{Z}) \overset{a.s.}{=} \int \frac{d\mu_{\mathcal{Z}}(t)}{w^2 - t^2} \delta_{i,j}$$

$$Q_{i,j}(w \mid \mathcal{V}, \mathcal{Z}) \overset{a.s.}{=} \int \frac{d\mu_{\mathcal{Z}}(t)}{w^2 - t^2} \delta_{i,j}$$

where $\mu_{\mathcal{Z}}(t)$ is the density of the Marčenko-Pastur distribution [25] given by

$$\mu_{\mathcal{Z}}(t) = \frac{4\sigma^4\beta - (t^2 - \sigma^2 - \sigma^2\beta)^2}{2\pi\sigma^2\beta t} \mathbf{1}_{\beta_-, \beta_+}(t), \tag{17}$$

where $\beta_{\pm} = \sigma^2(1 \pm \sqrt{\beta})^2$ .

The next lemma shows that the matrix with entries $A_{i,j}X_{i,j}$, with $A_{i,j} \overset{iid}{\sim} (\mu_A, \sigma_A^2)$, and $X_n$ from our signal model, is well approximated by the matrix with entries $\mathbb{E}[A]X_{i,j}$.

**Lemma 5.** Let $A$ be a random variable with mean $\mu_A$ and variance $\sigma_A^2$. Let $\{X_n\}$ be a matrix sequence satisfying assumptions **A1–A3** (Section 2 in the main text). Let $\delta_{n,1}$ be the largest singular value of the matrix $\Delta_n$ with entries

$$(\Delta_n)_{i,j} = A_{i,j}X_{i,j} - \mu_A X_{i,j} .$$

Then $\delta_{n,1} \overset{a.s.}{\to} 0$ as $n \to \infty$.

*Proof.* See [27] equations no. 35–38. $\qquad\square$

The next lemma shows that adding a "small perturbation" to $Z_n$ does not change the value of $H$ and $Q$ from Definition 6.

**Lemma 6.** Let $\mathcal{Z} = \{Z_n\}_{n=0}^{\infty}$ a sequence of matrices s.t. $Z_n$ is $m_n$-by-$n$ and $\lim_{n\to\infty} m_n/n = \beta \leq 1$, and let $\mathcal{U}$ and $\mathcal{V}$ be arbitrary sequences of orthonormal matrices as in Definition 6, which are either nonrandom or independent of $\{Z_n\}$. Assume that for some $1 \leq i \leq m_n$ and $1 \leq j \leq n$,

$$H_{i,j}(w \mid \mathcal{U}, \mathcal{Z}) \overset{a.s.}{=} f(w)\delta_{i,j} \qquad \text{and}$$

$$Q_{i,j}(w \mid \mathcal{V}, \mathcal{Z}) \overset{a.s.}{=} f(w)\delta_{i,j} .$$

Let $\{\Delta_n\}_{n=0}^{\infty}$ be a sequece of matrices of the same sizes and assume that $\delta_{n,1} \overset{a.s.}{\to} 0$ as $n \to \infty$, where $\delta_{n,1}$ is the largest singular value of $\Delta_n$ ($n = 1, 2, \ldots$). Denote by $\bar{\mathcal{Z}}$ the sequence of matrices $\{Z_n + \Delta_n\}$. Then also

$$H_{i,j}(w \mid \mathcal{U}, \bar{\mathcal{Z}}) \overset{a.s.}{=} f(w)\delta_{i,j} \qquad \text{and}$$

$$Q_{i,j}(w \mid \mathcal{V}, \bar{\mathcal{Z}}) \overset{a.s.}{=} f(w)\delta_{i,j} .$$

*Proof.* See [27] equations no. 33 and 34. $\qquad\square$

The next lemma in this chain of arguments is due to [1]:

**Lemma 7.** Let $\mathcal{Z} = \{Z_n\}_{n=0}^{\infty}$ a sequence of matrices s.t. $Z_n$ is $m_n$-by-$n$ and $\lim_{n\to\infty} m_n/n = \beta \leq 1$. Assume that $X_n$ is a sequence of matrices satistfying assumptions **[A1]–[A3]** from Section 2. Define $Y_n = X_n + Z_n$ and let $X_n = \sum_{i=1}^{r} x_i \tilde{\mathbf{u}}_i \tilde{\mathbf{v}}_i'$ and $Y_n = \sum_{i=1}^{m_n} y_{n,i} \mathbf{u}_{n,i} \mathbf{v}_{n,i}'$ denote their singular value decompositions, respectively. Assume that for any $\mathcal{U}$ and $\mathcal{V}$, which are arbitrary sequences of orthonormal matrices as in Definition 6, either nonrandom or independent of $\{Z_n\}$, we have for all $1 \leq i \leq m_n$ and $1 \leq j \leq n$,

$$H_{i,j}(w \mid \mathcal{U}, \mathcal{Z}) \overset{a.s.}{=} \int \frac{d\mu_{\mathcal{Z}}(t)}{w^2 - t^2} \delta_{i,j}$$

$$Q_{i,j}(w \mid \mathcal{V}, \mathcal{Z}) \overset{a.s.}{=} \int \frac{d\mu_{\mathcal{Z}}(t)}{w^2 - t^2} \delta_{i,j} ,$$

where $\mu_{\mathcal{Z}}(t)$ is given by (17). Then

1. For each $1 \leq i \leq r$ we have

$$\lim_{n \to \infty} y_{n_i} \overset{a.s.}{=} \begin{cases} \sigma(\sqrt{(\frac{x_i}{\sigma} + \frac{\sigma}{x_i})(\frac{x_i}{\sigma} + \frac{\beta\sigma}{x_i})}) & \text{for } x_i > \sigma\beta^{\frac{1}{4}} \\ \sigma(1 + \sqrt{\beta}) & \text{for } x_i \leq \sigma\beta^{\frac{1}{4}}. \end{cases} \tag{18}$$

2. Let $1 \leq i \leq r$ and $1 \leq j \leq m_n$. If $x_i > \sigma_B\beta^{\frac{1}{4}}$ and $j \leq r$, we have

$$d \cdot \lim_{n \to \infty} |\langle u_{n_i}^0, u_{n_j} \rangle|^2 \overset{a.s.}{=} \begin{cases} \dfrac{(\frac{x_i}{\sigma})^4 - \beta}{(\frac{x_i}{\sigma})^4 + \beta(\frac{x_i}{\sigma})^2} & x_i = x_j \\ 0 & x_i \neq x_j \end{cases} \tag{19}$$

and

$$d \cdot \lim_{n \to \infty} |\langle v_{n_i}^0, v_{n_j} \rangle|^2 \overset{a.s.}{=} \begin{cases} \dfrac{(\frac{x_i}{\sigma})^4 - \beta}{(\frac{x_i}{\sigma})^4 + (\frac{x_i}{\sigma})^2} & x_i = x_j \\ 0 & x_i \neq x_j \end{cases} \tag{20}$$

Otherwise, both quantities converge a.s to 0 as $n \to \infty$.

We can finally connect the dots and prove Lemma 2.

**Proof of Lemma 2.** . Per the lemma statement, let $Y_n = f_{A_n, B_n}(X_n)$ be a matrix sequence following our signal model (Definition 2) with $A_n \overset{iid}{\sim} (\mu_A, \sigma_A^2/n)$ and $B_n \overset{iid}{\sim} (0, \sigma_B^2/n)$. Let $\mathbf{A}_n$ be an $m_n$-by-$n$ matrix with entries $(\mathbf{A}_n)_{i,j} \overset{iid}{\sim} A_n$ and let $\mathbf{B}n$ be an $m_n$-by-$n$ matrix with entries $(\mathbf{B}_n)_{i,j} \overset{iid}{\sim} B_n$. We can write $(Y_n)_{i,j} = (\mathbf{A}_n)_{i,j}(X_n)_{i,j} + (\mathbf{B}_n)_{i,j}$. Letting $\Delta_n$ be the $m_n$-by-$n$ matrix with entries $(\Delta_n)_{i,j} = (\mathbf{A}_n)_{i,j}(X_n)_{i,j} - \mu_A(X_n)_{i,j}$ we have

$$Y_n = \mu_A X_n + \mathbf{B}_n + \Delta_n.$$

By Lemma 5, the top singular value $\delta_{n,1}$ of $\Delta_n$ satisfies $\delta_{n,1} \overset{a.s.}{\to} 0$ as $n \to \infty$.

Now, choose arbitrary sequences of orthonormal matrices $\mathcal{U}$ and $\mathcal{V}$ as in Definition 6, either nonrandom or independent of $\{\mathbf{B}_n\}$. Let $\mathcal{B}$ denote the matrix sequence $\{\mathbf{B}_n\}$ and let $\bar{\mathcal{B}}$ denote the matrix sequence $\{\mathbf{B}_n + \Delta_n\}$. Invoking Lemma 6 we obtain

$$H_{i,j}(w \,|\, \mathcal{U}, \mathcal{B}) \overset{a.s.}{=} H_{i,j}(w \,|\, \mathcal{U}, \bar{\mathcal{B}})$$
$$Q_{i,j}(w \,|\, \mathcal{V}, \mathcal{B}) \overset{a.s.}{=} Q_{i,j}(w \,|\, \mathcal{V}, \bar{\mathcal{B}}).$$

for all $1 \leq i \leq m_n$ and $1 \leq j \leq n$. However, by Lemma 4, for all $1 \leq i \leq m_n$ and $1 \leq j \leq n$ we have

$$H_{i,j}(w \,|\, \mathcal{U}, \mathcal{B}) \overset{a.s.}{=} \int \frac{d\mu_{\mathcal{Z}}(t)}{w^2 - t^2} \delta_{i,j}$$
$$Q_{i,j}(w \,|\, \mathcal{V}, \mathcal{B}) \overset{a.s.}{=} \int \frac{d\mu_{\mathcal{Z}}(t)}{w^2 - t^2} \delta_{i,j},$$

where $\mu_{\mathcal{Z}}(t)$ is given by (17) with $\sigma \equiv \sigma_B$. It follows that the sequence $Y_n = \mu_A X_n + (\mathbf{B}_n + \Delta_n)$ satisfies all the assumptions of Lemma 7, for arbitrary $\mathcal{U}$ and $\mathcal{V}$. We therefore conlude that, for $Y_n$, equations (18), (19) and (20) hold, replacing $\sigma$ with $\sigma_B$ and $x_i$ with $\bar{x}_i$ ($1 \leq i \leq r$). The lemma follows. $\square$

# 9 Contamination modes

We describe here contamination modes that did not enter the main text.

**Basic contamination modes**

1. **Additive noise.** The simplest form of contamination is noise added to each entry. Assume $Y_{i,j} = X_{i,j} + Z_{i,j}$. The matching parameters for the model $A \odot X + B$ are $\mu_A = 1, \sigma_A = 0, \sigma_B = \sigma_Z$.

2. **Missing-at-random.** Assume data with entries that are missing-at-random, where missing entries are replaced by zeros with probability (w.p.) $1 - \kappa$. Assume

$$Y_{i,j} = \begin{cases} X_{i,j} & \text{w.p. } \kappa \\ 0 & \text{w.p. } 1 - \kappa \end{cases},$$

.

   The matching parameters for the model $A \odot X + B$ are $\mu_A = \kappa, \sigma_A = 0, \sigma_B = 0$.

3. **Outliers-at-random.** When some entries of $Y$ contain inordinate level of noise, the corresponding entries are said to be outliers:
   Assuming that this noise is additive, we can write

$$Y_{i,j} = \begin{cases} X_{i,j} & \text{w.p. } \kappa \\ X_{i,j} + W_{i,j} & \text{w.p. } 1 - \kappa \end{cases}.$$

   With $W_{i,j} \overset{\text{iid}}{\sim} (0, \tau^2)$. The matching parameters for the model $A \odot X + B$ are $\mu_A = 1, \sigma_A = 0, \sigma_B = \sqrt{(1 - \kappa)\tau^2}$.

4. **Multiplicative Noise.** Each signal entry is multiplied by a random noise distribution sample. $Y_{i,j} = Z_{i,j} \cdot X_{i,j}$. The matching parameters for the model $A \odot X + B$ are $\mu_A = \mu_Z, \sigma_A = \sigma_Z, \sigma_B = 0$.

5. **Corrupt-at-random.** In some measurement processes some entries are completely destroyed and *replaced* by randomly generated noise. To distinguish this form of contamination from simple outliers, where the original entry is not replaced, we refer to this as corruption. Assume

$$Y_{i,j} = \begin{cases} X_{i,j} & \text{w.p. } \kappa \\ W_{i,j} & \text{w.p. } 1 - \kappa \end{cases}.$$

   ,with $W_{i,j} \overset{\text{iid}}{\sim} (0, \tau^2)$. The matching parameters for the model $A \odot X + B$ are $\mu_A = \kappa, \sigma_A = 0, \sigma_B = \sqrt{(1 - \kappa)\tau^2}$.

**Composite contamination modes**

1. **Additive noise and missing-at-random.** Assume that additive noise has been added and then some entries were deleted and replaced with zeros. Then

$$Y_{i,j} = \begin{cases} X_{i,j} + Z_{i,j} & \text{w.p. } \kappa \\ 0 & \text{w.p. } 1 - \kappa \end{cases}.$$

   The equivalent $A$,$B$ selection is: $\mu_A = \kappa, \sigma_A = 0, \sigma_B = \sqrt{\kappa\sigma_Z^2}$.

2. **Additive noise and outliers-at-random.**

$$Y_{i,j} = \begin{cases} X_{i,j} + Z_{i,j} & \text{w.p. } \kappa \\ X_{i,j} + W_{i,j} & \text{w.p. } 1 - \kappa \end{cases}.$$

   Where $W_{i,j} \overset{\text{iid}}{\sim} (0, \tau^2)$, and $\tau$ is large compared to $\sigma_Z$. The equivalent $A$,$B$ selection is: $\mu_A = 1, \sigma_A = 0, \sigma_B = \sqrt{\kappa\sigma^2 + (1 - \kappa)\tau^2}$.

3. **Multiplicative noise and corrupt-at-random.**

$$Y_{i,j} = \begin{cases} Z_{i,j}X_{i,j} & \text{w.p. } \kappa \\ W_{i,j} & \text{w.p. } 1 - \kappa \end{cases}.$$

   The equivalent $A$,$B$ selection is: $\mu_A = \kappa \cdot \mu_A, \sigma_A = 0, \sigma_B = \sqrt{(1 - \kappa)\tau^2}$.

## 10 Conclusions

Singular value shrinkage emerges as an effective method to reconstruct low-rank matrices from contaminated data that is both practical and well understood. Through simple, carefully designed manipulation of the data singular values, we obtain an appealing improvement in the reconstruction mean square error. While beyond our present scope, following [16], it is highly likely that the optimal shrinker we have developed offers the same mean square error, asymptotically, as the best rotation-invariant estimator based on the data, making it asymptotically the best SVD-based estimator for the target matrix.

## Acknowledgements

DB was supported by Israeli Science Foundation grant no. 1523/16 and German-Israeli Foundation for scientific research and development program no. I-1100-407.1-2015.

## Footnotes

[1]The incoherence assumption is widely used in related literature [6, 12, 27], and asserts that the singular vectors are spread out so $X$ is not sparse and does not share singular subspaces with the noise.

[2]The full Matlab code that generated the figures in this paper and in the Supporting Information is permanently available at `https://purl.stanford.edu/kp113fq0838`.