[Reviews · NeurIPS 2017]

Reviewer 1



This paper consider the problem of matrix de-noising under the mean square-loss via estimators that shrink the singular values of a contaminated version of the matrix to be estimated. Specifically, the authors seek asymptotically optimal shrinkers under various random contamination models, extending beyond additive white noise. Unfortunately, I could not determine the novelty of the techniques used as there is no clear indication in the paper of what results are new or rather simple extensions of previous results, such as in Nadakuditi (2014) and Gavish & Donoho (2017). As a non-expert, as far as I can see, the results in this paper are rather simple extensions of those given in Gavish and Donoho. A clear indication of what is new and a detailed comparison to what is already known is needed. In addition, I found many sentences to be unclear and inadequate, making it a difficult read. Some more specific questions/comments: In line 213 it is not clear whether you prove that the a.s. limit exists or assume it exists. It is unclear under what class of estimators is yours optimal. Only a clue is given in the very last sentence of the paper. In line 64, the variance of A's entries is taken to be the same as that of B, namely \sigma_B, while later it is changed to sigma_A. Thm. 2 however does not involve sigma_A. Is there a specific reason for choosing sigma_B = sigma_A? In addition, it is unclear how can one estimate mu_A, sigma_A and sigma_B, which are needed for the method. Lines 89 and 90 should come before Thm. 1. The hard threshold shrinker is missing a multiplicative y_i all over the paper. There are many typos.

Reviewer 2



This paper introduces a new model of i.i.d. noise in low-rank matrices, which subsumes additive and multiplicative noise, corruption, outliers, and missing entries. The paper derives exact optimal shrinkage and thresholding for this noise model, and demonstrates the results empirically. This unification is an improvement over previous literature in which a variety of specific noise models are studied; the model and analysis here make a nice contribution to the field. Indeed, this model could be of interest in other questions of low-rank recovery, such as sparse or non-negative PCA. A more detailed overview of prior work in singular value shrinkage would help the introduction.

Reviewer 3



his paper is about how to design an optimal shrinkage function such that the latent low-rank matrix can be best reconstructed from the singular vectors of an observation matrix. Unlike existing works, which mainly focus on the additive white noise model, this paper studies a general contamination model that can involve various noise regimes such as missing data and outliers. The authors established a set of theorems that give us some useful messages, e.g., the optimal shrinkage function is unique and determined explicitly. Overall, I think this is a solid work. However, it is unclear how the proved results can take effect in practice. The optimal shrinkage function depends on some extrinsic parameters of the contamination model, which is unknown. So, one may need to estimate those parameters based on the observations. But in this case there is no guarantee for the optimality of the shrinkage function. It would be better if the authors could test their results on some realistic datasets. Moreover, the papers contain several typos. Please proof-read the paper.